# VLAC: A Generalist Action-Critic Model via Pair-wise Progress Understanding

## Abstract

Recent advances in Vision-Language-Action (VLA) models have significantly improved robotic perception and manipulation capabilities. However, robots deployed in real-world settings still struggle to adapt in dynamic, open-ended environments due to a lack of reliable task progress feedback and improvement mechanisms. To address these challenges, we propose a generalist Vision Language Action-Critic model, VLAC, which can integrate both human and robot data, and unify action generation and task progress understanding within a single autoregressive architecture. Specifically, we propose a scalable and generalizable pair-wise progress understanding approach to predict the task progress delta between any two images in one visual trajectory, and generate the action based on the first image. The model is trained on large-scale, multi-source human data without action annotations and robot data with action information, while also incorporating general vision-language data yielding world knowledge understanding. Furthermore, we deploy reinforcement learning where VLAC can autonomously evaluate task progress to feedback intrinsic rewards. We evaluated our model's progress understanding across eight datasets and show that it not only generalizes to new tasks and environments but also discriminates success from failure trajectories, e.g., on RoboFAC dataset, it reaches VOC-F1 0.89 for successful versus 0.44 for failed trajectories, providing dependable dense reward signals. Then, we evaluated action generation and real-world reinforcement learning performance on diverse real-world robotic manipulation tasks. Experimental results indicate strong disturbance robustness in VLAC's action generation, while integrating pairwise progress prediction allows real-world RL to improve success from roughly 30% to 90% within 200 episodes.

## 1 Introduction

With the rapid development of Vision-Language-Action (VLA) models, the intelligence of robotic perception and manipulation capabilities has greatly improved, leading to impressive performance in autonomously completing general tasks. Current VLA models are primarily trained through imitation learning, which requires vast amounts of data Lin et al. (2024); Team et al. (2025); Deng et al. (2025); Bjorck et al. (2025). However, collecting human expert trajectories is not only costly and time-consuming, but most data collection efforts focus on laboratory-customized scenes and tasks. Consequently, significant barriers remain for robots to perform effectively in dynamic real-world scenarios, particularly concerning cross-scene generalization and robustness. Moreover, these models lack efficient feedback mechanisms for learning and improvement in new scenarios.

To address this limitation, action models with strong generalization and critic models that distinguish desirable from undesirable behavior can jointly acquire visuomotor skills in new real-world environments Ma et al. (2024). In real-world learning, reinforcement learning (RL) enables autonomous exploration via critic feedback, yet many methods still depend on hand-crafted, task-specific shaping Mendonca et al. (2024); Herzog et al. (2023); Mendonca et al. (2023); Kumar et al. (2024); Xu et al. (2022). Approaches advertised as reusable often require additional task-dependent data to train reward surrogates or termination classifiers Luo et al. (2024); Hu et al. (2023), and sparse signals usually appear only near completion, leaving intermediate progress unscored. Although some works propose denser or universal progress signals Ma et al. (2023; 2022), they generalize poorly to unseen tasks, objects, or goal language. Recent work uses pretrained Vision-Language Models (VLMs) for progress estimation Ma et al. (2024), yet evaluation usually relies on single-point estimates, lacks fine-grained continuous value modeling, and depends solely on expert trajectories,

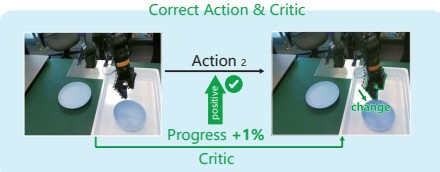

Figure 1: Examples of robot manipulation and progress prediction.

restricting scene and task diversity. In parallel, VLA models built on VLM pretraining aim to improve manipulation generalization Black et al. (2024); Intelligence et al. (2025); Team et al. (2025); Kim et al. (2024), but training is typically confined to end-to-end action generation plus generic image understanding, without explicit modeling of task process understanding or adaptive mechanisms for new environments. Achieving fine-grained, precise, and generalizable critic feedback and integrating it with action generation to enable broad task manipulation generalization thus remains challenging.

To this end, in this paper, we propose a generalist Vision-Language-Action-Critic model, VLAC, which unifies the roles of "actor" and "critic" within a single autoregressive architecture via pair-wise progress understanding, capable of dense and precise task progress prediction and robust action generation. Some real-world interaction examples of robot manipulation and progress prediction are shown in Figure 1. In order to provide dense and precise feedback signals that effectively reflect the quality of robotic actions and can be scaled to large-scale unlabeled data, we propose an image observation pair-wise progress prediction method. This approach leverages visual images to assess the relative task progress between two states, focusing on the differences between image observations. It is capable of detecting subtle changes and learning variations across different temporal scales, making it adaptable to various task lengths and scenarios. During training, annotations and negative samples are automatically generated based on temporal relationships, enabling convenient large-scale expansion. In addition to progress prediction, auxiliary tasks such as object detection, 3D scene understanding, and pair-wise image differences detection are introduced to further enhance the critic's capabilities. We also constructed an in-context learning dataset, which includes examples of similar tasks along with corresponding progress labels.

Then, the VLAC model was trained to simultaneously handle action generation and progress understanding. When the dataset contains action data, we construct corresponding action train data aligned with progress understanding, enabling VLAC to generate the appropriate actions based on the first image of the pair mentioned above. We connect the actor and critic tasks through shared states, enabling the model to understand both the process and the action generation at each stage. The pair-wise progress understanding further provides insights into changes before and after actions, enhancing the model's comprehension of action effects. For action output, we use the delta end-of-effector (EEF) position as action and experiment with both semantic discrete actions and FAST Pertsch et al. (2025) discrete actions. This approach offers improved spatial awareness, better alignment with semantic space, fully leverages process understanding, and delivers stronger generalization capabilities.

We conduct comprehensive evaluations of VLAC across eight public datasets (six unseen during training) and observe strong cross-task, cross-entity generalization, further enhanced via in-context learning; notably, it attains a VOC-F1 score of 0.89 on successful processes versus 0.44 on failed ones, enabling fine-grained dataset quality estimation. For action generation, we assess VLAC on diverse real-world manipulation tasks, where it consistently generates plausible actions and maintains robustness under extreme lighting variation. In real-world reinforcement learning, VLAC adaptively leverages both successful and failed trajectories with progress-based rewards, autonomously improving its success rate from roughly 30% to 90% within 200 episodes.

In summary, our contributions are:

1. Vision-Language-Action-Critic Model, a generalist action and critic unified model, which has open-ended general task progress prediction and robust action generation capabilities.

2. Pair-wise progress understanding method modeling the relative delta changes between pairs of images in visual trajectories, providing fine-grained, accurate, and generalizable results.

3. VLAC is jointly trained on action-free human data, web videos, and action-labeled robot datasets. It is evaluated on eight datasets (six unseen) for critic capability, and on real-world manipulation tasks for action generation performance and on-robot self-improvement.

## 2 RELATED WORK

**Embodied reward models.** Some studies consider referencing different data sources to learn transferable reward and value functions to guide manipulation tasks. Early researches use various forms of data as criteria, such as robot Sermanet et al. (2016) human videos with discriminators Dasari & Gupta (2021) and multi-player arenas Sun et al. (2025), etc. Some of them also design algorithms to learn functions, including contrastive learning Zitkovich et al. (2023) and offline RL methods Ma et al. (2022; 2023); Bhateja et al. (2023). Recently, with the rise of LLMs and VLMs, research on utilizing them as reward models has begun to emerge. To leverage their encoding abilities, some works utilize them to learn an embedding computing function and calculate current observations and goals into embedding vectors, and take their distances as reward Zhang et al.; Ma et al. (2022; 2023); Bhateja et al. (2023). Some methods Zhou et al. (2024); Li et al. (2025) further integrate encoding and generating abilities, which synthesize goal observations and regard discrepancies between current observations and them as the progressive value. Furthermore, some models treat reward and value function learning as related tasks to optimize the training objectives. For instance, Chen et al. (2025) formulates reward modeling as reasoning tasks. Besides these indirect progress modeling methods, recent works directly predict a concrete number of current progress based on input observations, such as Ma et al. (2024). Despite achieving significant breakthroughs, existing literatures are mainly based on single-point estimates, thus lacking the ability of comprehensive progress understanding and continuous reward esimating. Our proposed VLAC is a mechanism of pair-wise progress understanding to tackle this problem.

**Vision-Language-Action Models (VLAs).** VLAs integrate the capabilities of visual analysis, language understanding, and action prediction, demonstrating powerful performances in robot manipulation tasks. Early works including RT-1 Brohan et al. (2022) and RT-2 Zitkovich et al. (2023) unlock end-to-end policy learning by action tokenization. After that, open-source approaches such as Octo Team et al. (2024), OpenVLA Kim et al. (2024), and OpenVLA-OFT Kim et al. (2025) have emerged. Flow matching-based methods, $\pi_0$ Black et al. (2024) and $\pi_{0.5}$ Intelligence et al. (2025), for instance, also verify the power of this paradigm. Recently, much research on VLA has focused on progress understanding and wrong action detection. For example, CoT-VLA Zhao et al. (2025) applies a chain of thought to split a task into progressive subtask instructions to improve VLA task understanding. FPC-VLA Yang et al. (2025) utilizes a fine-tuned supervisor to check failure in the manipulation process. These works validate the crucial meaning of reward modeling for progress on robot manipulation.

## 3 VISION-LANGUAGE-ACTION-CRITIC MODEL

We constructed a Vision-Language-Action-Critic (VLAC) model based on pair-wise task progress understanding, enabling both action generation and delta task progress prediction (Figure 2). VLAC leverages diverse data sources: public robotic manipulation datasets, human demonstration data, our self-collected manipulation data, and general image understanding datasets. Video trajectories are converted into pair-wise samples to learn relative task progress between arbitrary states, with each pair augmented by a task description and task completion evaluation to enable task progress estimation and action generation (left of the figure). The pair-wise formulation is agnostic to data collection strategies and segment starting points, improving robustness and generality. For each pair, the model outputs a delta progress value: a positive value indicates that the second image corresponds to a more advanced stage of task completion, while a negative value indicates regression. VLAC can estimate task progress and identify failing actions or trajectories, providing dense reward signals for real-world reinforcement learning and guiding data refinement. It can also operate as a Vision-Language-Action model to directly perform manipulation, exhibiting zero-shot generalization to varied scenarios (right of the figure). In the following, we detail four subsections: Pair-wise Task Progress Understanding, Semantic Space Action Generation, General Vision-Language Understanding, and Multi-Source Training Datasets.

### 3.1 PAIR-WISE TASK PROGRESS UNDERSTANDING

Humans often encounter limitations in their initial capabilities when faced with new environments and new tasks. However, their ability to understand the progression of the task is exhibited in a higher generalizability than their ability to execute tasks. This ability enables continuous assessment of the task progress across various processes, thereby optimizing one's actions and achieving sustained improvement. Rich general knowledge thus endows the understanding of task progress with strong

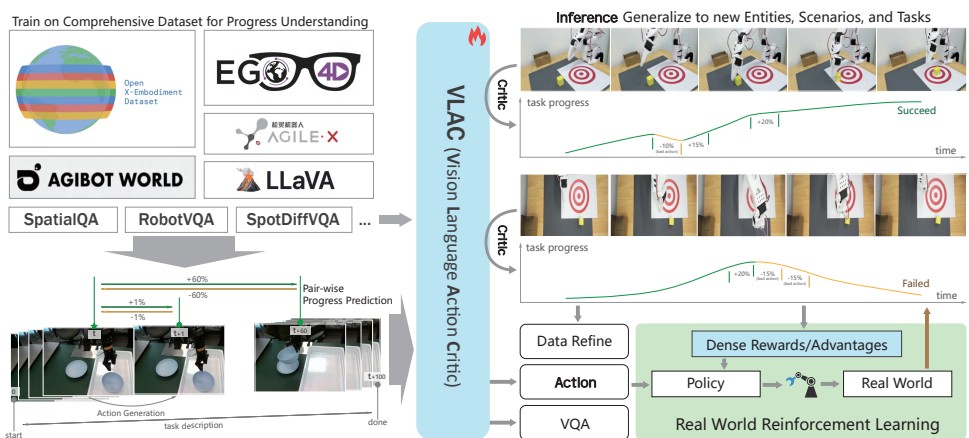

Figure 2: Overall framework of the VLAC model. The left panel illustrates the training stage, where pair-wise progress understanding is optimized. The right panel shows the inference stage for progress estimation and action generation.

generality, which can facilitate capabilities when faced with different environments and tasks. Inspired by this, for robot manipulation models to be deployed in the complexity of the real world and execute general tasks, they must not only understand language and visual information, but also understand task progress, which can indicate changes in task completion status across different processes. We design a pair-wise task progress understanding method that is unaffected by the initial point of the task. This method integrates general human data without action information and robot data annotated with actions to estimate fine-grained variations in task processes robustly.

We formalize the task process as a video segment with description $V = (O, l_{\text{task}})$, which consists of a basic RGB image sequence $O = (o_1, \ldots, o_T)$ and a textual task goal description $l_{\text{task}}$. Each training trajectory in our dataset is a successful and efficient execution of the annotated task. To understand the variations of the task process, we assume that task progress is positively correlated with time; as time increases, the task progresses. Thus, pair-wise task progress understanding can be formalized as

$$c_{i,i+\Delta t} = \text{VLAC}(o_i, o_{i+\Delta t}; l_{\text{task}}), \tag{1}$$

where $\Delta t$ denotes the time difference between two frames, and $c_{i,i+\Delta t}$ represents the degree to which the task progresses in $o_{i+\Delta t}$ advances the task relative to $o_i$. Specifically, $\Delta t \in [-i+1, T-i] \cap \mathbb{Z}$, which allows us to focus on both fine-grained single-step changes and long-term task progress, mitigating noise from minor variations while naturally constructing balanced negative samples. During training, $c_{i,i+\Delta t}$ is annotated according to the natural temporal order as $c_{i,i+\Delta t} = \Delta t/(T-i)$, representing the percentage of task progress from $o_i$ to $o_{i+\Delta t}$. This task progress understanding approach is independent of data collection strategies and starting points of segment, enhancing generality and robustness, as illustrated in the lower left corner of Figure 2.

To further enhance the semantic understanding of the task process, we construct a task description estimation objective:

$$\hat{l}_{\text{task}} = \text{VLAC}(o_{i_{\text{start}}}, o_{i_{\text{end}}}), \tag{2}$$

where $o_{i_{\text{start}}} \in [0, 0.3T] \cap \mathbb{Z}$ and $o_{i_{\text{end}}} \in [0.8T, T] \cap \mathbb{Z}$. By generating task descriptions from the initial and final frames, the task description becomes not only an input condition but also an output target, thereby improving the joint understanding of vision and language.

To enhance understanding of task completion, we design a task completion judgment task:

$$l_{\text{done}} = \text{VLAC}(o_i; l_{\text{task}}), \tag{3}$$

where if $i < 0.8T$, $l_{\text{done}} = 0$ indicates the task is not yet completed, and if $i > 0.95T$, $l_{\text{done}} = 1$ indicates the task is completed. Considering the diversity of data and collection strategies, it is difficult to accurately determine the exact completion point; thus, for $0.8T \leq i \leq 0.95T$, no training label is made to ensure label accuracy. By learning to judge task completion, the model's understanding of completion conditions is enhanced, providing auxiliary signals for task completion in real-world reinforcement learning.

We use different prompts to distinguish different tasks. Furthermore, to improve the task progress understanding, we design four data construction strategies:

1. Pair-wise Image Difference Filtering: We assume task progress is positively correlated with time, which generally holds but may be violated in noisy data or segments with minimal change (e.g., static scenes). To mitigate the impact of such noise, we set the interval between $i$ and $i + 1$ to approximately 0.2s during data construction and compare the pixel difference between the two frames. If $\mathrm{Diff}(o_i, o_{i+\Delta t}) < \sigma$, then set $c_{i,i+\Delta t} = 0$, indicating the two frames are in the same progress. In our experiments, we set $\sigma = 1\%$, enabling the model to focus on significant changes and improving the robustness of task progress understanding.

2. Pair-wise Progress Understanding with Joint Sampling: Inspired by contrastive learning, to ensure data balance and symmetry between forward and reverse processes, for each sampled pair $(o_i, o_{i+\Delta t})$, we construct four related data samples as a mini group within a batch:

$$\{(o_i, o_{i+1}), (o_{i+1}, o_i), (o_i, o_{i+\Delta t}), (o_{i+\Delta_t}, o_i)\}. \tag{4}$$

This covers both forward and backward directions, as well as fine-grained and global understanding, as shown in the lower left of Figure 2.

3. Task Completion Judgment Joint Sampling: For data balance in the task completion judgment task, each time we sample a pair of data: one from a completed state and one from an incomplete state within the same trajectory.

4. Cross-sampling of Task Descriptions and Image Sequences: To enhance the model's ability to distinguish whether a process matches the task description, we sample, with a 5% probability during pair-wise data sampling, a task description $l_{\text{task}}$ that does not belong to the current trajectory, setting $c_{i,i+\Delta t} = 0$. This method aims to improve the alignment of semantic and progress understanding of the model.

Cross-scene and cross-task transferability remains a key challenge for embodied intelligence models on the path to generalization. When humans adapt to new environments and tasks, their initial capabilities may also be limited; however, having reference examples can significantly improve both initial performance and learning efficiency. Inspired by this, and to improve the cross-scene and cross-task transferability of VLAC, we further enhance progress understanding with in-context learning, enabling effective learning from a single reference example. Specifically, in-context progress understanding can be formalized as

$$c_{i,i+\Delta t} = \mathrm{VLAC}(o_i, o_{i+\Delta t}; l_{\text{task}}, O_{\text{ref}}, o_0), \tag{5}$$

where $O_{\text{ref}}$ is the reference process, which may be provided by a robot demonstration or a human demonstration, offering guidance on both scene and task logic. $o_0$ is the starting point of the current trajectory and can be optionally included as input, enhancing the model's ability to align with the reference process and enabling inference of the absolute progress of $o_i$ and $o_{i+\Delta t}$.

These tasks we construct do not require action information, thereby avoiding the issue of inconsistent action spaces across entities. This design also allows our approach to apply to both human and robot data, enabling the use of large-scale, diverse human data to significantly improve model generalization and alleviate the scarcity of robot data in the real world. Moreover, in-context learning endows the model with rapid transfer capabilities and further enhances its generalization.

### 3.2 SEMANTIC SPACE ACTION GENERATION

Based on general task process understanding, we further construct an action generation task in the semantic space to achieve multi-task control of a robotic arm. Since the general generation capabilities of pretrained multimodal models are mainly in the semantic space, and task understanding is also generated in the semantic space, we fully leverage this knowledge and the strong semantic representations of pretrained models by representing actions as numbers and generating them in the semantic space by the autoregressive approach.

To further improve spatial reasoning performance, the action is represented as the delta End-Effector (eef) pose, which is a general spatial representation while remaining independent of embodied entities:

$$a_i = \mathrm{VLAC}(o_i^0, \ldots, o_i^k; s_i; l_{\text{task}}),$$

where $o_i^k$ denotes the $k$-th viewpoint at the $i$-th step, $s_i$ represents the state of the robotic arm, and $a_i$ is the action to be executed at the $i$-th step, represented as a string of numbers. Meanwhile, $o_i^k$ and the

subsequent image $o_{i+1}^k$, obtained after executing action $a_i$, can be paired for progress understanding. Additionally, we also experiment with FAST Pertsch et al. (2025) as the action tokenizer for action chunk output.

With this formulation, VLA demonstrates strong semantic and scene generalization capabilities. Moreover, the generated actions can be sampled with diversity within a reasonable range, which is beneficial for exploration and improvement in reinforcement learning.

### 3.3 General vision language understanding

To enhance the model's multimodal understanding capabilities, we incorporate a series of publicly available VQA datasets, focusing on four aspects: general conversational ability, robotic understanding, spatial reasoning, and pair-wise image difference distinction. Specifically, we select the following datasets:

LLAVA Liu et al. (2023): This dataset includes basic multi-turn dialogues and a rich collection of VQA data, which helps maintain the model's general multimodal understanding and conversational abilities.

RoboVQA Sermanet et al. (2024): This dataset contains VQA data from various robotic tasks, aimed at improving the model's understanding of multimodal data in robotic scenarios.

SpatialVQA Chen et al. (2024): This dataset provides spatial reasoning data based on RGB images, including depth estimation, object detection, and more, which strengthens the model's ability to understand and estimate spatial information within images.

Spot-the-diff Jhamtani & Berg-Kirkpatrick (2018), InstructPix2Pix Brooks et al. (2023): These datasets consist of pair-wise image difference comparison data, designed to enhance the model's capability to detect fine-grained differences between images, thereby supporting pair-wise progress understanding tasks.

By leveraging these datasets, we aim to comprehensively improve the model's multimodal reasoning and understanding capabilities across diverse application domains.

### 3.4 Multi-source train datasets

We train VLAC on a unified multi-source dataset spanning human real-world interaction video (Ego4D HOD Pei et al. (2025)), general vision-language instruction, editing and VQA style supervision (InstructPix2Pix Brooks et al. (2023), RobotVQA Sermanet et al. (2024), Spot the Diff Jhamtani & Berg-Kirkpatrick (2018), LLaVA Liu et al. (2023), SpatialQA Chen et al. (2024)), and diverse real-world robotic manipulation trajectories (AGIBOT Bu et al. (2025), Bridge Walke et al. (2023), DROID Khazatsky et al. (2024), FMB Luo et al. (2025), RoboSet Bharadhwaj et al. (2024)) plus our self-collected data, totaling roughly 40M sampled training instances (including a subset with multi-turn dialogue annotations) from about 4.2K hours of embodied and human interaction video, enabling joint modeling of pair-wise progress deltas, semantic grounding, and action generation. Full composition statistics and details are provided in the Appendix A.

## 4 Experiments

### 4.1 Implementation Details

During the pre-training phase of the VLAC model, we used a batch size of 3200 and set the maximum learning rate to 8e-4. The training was conducted on 200 A100-SXM4-80GB GPUs. The training time for the 8B model was approximately 11 days, while the 2B model required around 5 days. The robot in our real-world experiment is AGILE PiPER and it is controlled via a 7-DOF end effector based on the delta pose mechanism. We use the PPO algorithm to conduct the RL experiments.

### 4.2 Evaluation Datasets

To evaluate our VLAC model to understand task progress, especially its generalization to out-of-distribution scenarios such as new scenes, new tasks, and new entities, we conducted tests not only on the test sets included in the training datasets, but also on six additional datasets that were not seen during training. These include both human operation datasets and datasets containing failure processes. Specifically, in addition to the Bridge Walke et al. (2023) and Droid Khazatsky et al. (2024) datasets from the training set, we selected RT1 Brohan et al. (2022), RoboNet Dasari et al.

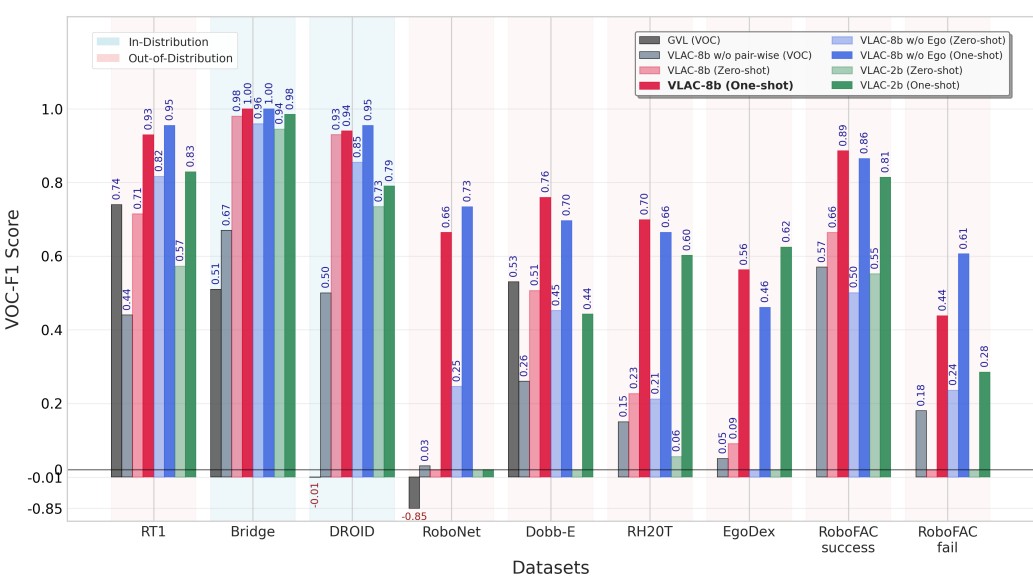

Figure 3: Progress Understanding Performance Comparison Across Different Models and Datasets.

(2019), Dobb-E Shafiullah et al. (2023), RH20T Fang et al. (2023), EgoDex Hoque et al. (2025), and RoboFAC Lu et al. (2025) for evaluation.

Dobb-E features a unique gripper and only provides gripper-perspective views, making it suitable for testing cross-entity and cross-viewpoint generalization. RoboNet lacks language annotations and does not exhibit smooth temporal structure, so it should show poor task and temporal correlation in the absence of reference examples. EgoDex is a human hand manipulation dataset, which can be used to evaluate general task process understanding and the model's compatibility with dexterous hand tasks. RoboFAC contains two subsets: one with successful task processes and one with failed processes, providing a direct way to evaluate the model's ability of progress understanding.

### 4.3 EVALUATION METRICS

Following GVL Ma et al. (2024), we use Value-Order Correlation (VOC) as the primary metric for task progress understanding. VOC is the rank correlation between predicted step values and their chronological order in an expert video for the single-point prediction method; higher values (-1 to 1) indicate monotonic task progression. To assess the robustness of pair-wise methods, we introduce Value-Reversed-Order Correlation (VROC), computed after reversing the sequence; performance is more stable when VROC is close to VOC. We report VOC-F1, the harmonic mean of VOC and VROC, to jointly capture forward and reversed consistency, which is better suited to pair-wise methods. For action generation, following $\pi_{0.5}$ Intelligence et al. (2025), we measure success rate and human-rated task progress, the latter enabling finer-grained evaluation when tasks are partially completed. In real-world RL experiments, we focus on success rate, reported as the moving average over the most recent 20 episodes. The detailed description of Evaluation Metrics can be seen in Appendix B.

### 4.4 VLAC CRITIC PERFORMANCE

The VLAC model trained on public robotic manipulation data mainly includes bridge, droid, roboset, fmb, AgiBot World, excluding the RT1 and other datasets, and the robotic arm entities, scenes, and tasks in these datasets are almost different from the above datasets. We will conduct further large-scale validation on these datasets. We trained three models: the VLAC-8b model with 8 billion parameters, the VLAC-8b w/o Ego model (which was trained without the Ego4D dataset), and the VLAC-2b model with 2 billion parameters. The overall performance across the eight datasets is shown in Figure 3. Compared to the GVL model, which is based on Gemini-1.5-pro and represents the state-of-the-art, VLAC demonstrates significant improvements in performance. To further validate the effectiveness of the pair-wise approach, we trained and tested VLAC-8B w/o pair-wise using a single-point prediction method similar to GVL. The results, as shown in the figure, indicate that the pair-wise method substantially enhances progress understanding. Our model demonstrates strong results on in-distribution datasets (Bridge and Droid), as well as robust generalization on out-of-

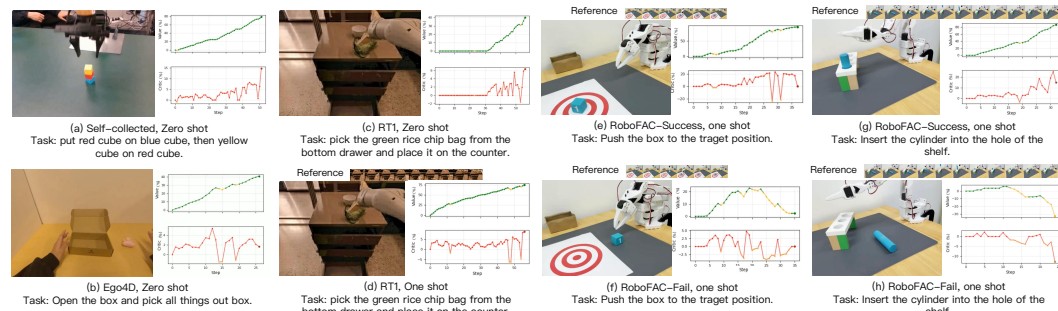

Figure 4: Example results of VLAC for task progress understanding.

distribution datasets. Notably, under one-shot conditions, the model's powerful contextual reasoning significantly improves its ability to assess task progression. For example, in the RT1 dataset, where the tasks, scenes, and robotic arms differ greatly from those in the training data, the VOC-F1 still reaches 0.95, indicating highly accurate task process prediction. On the RoboNet dataset, which lacks language annotations and does not exhibit smooth temporal structure, the VOC-F1 is 0 in the zero-shot setting—reflecting the model's correct reliance on language descriptions for task progression. However, when provided with examples, the one-shot performance improves dramatically, further highlighting the model's strong contextual learning capabilities. For the EgoDex dataset, which consists of human demonstration data, we observe that even without incorporating the Ego4D dataset, the model can still leverage context to understand task processes. After including Ego4D in training, this capability is significantly enhanced. The RoboFAC dataset contains both successful and failed task executions. Our method clearly distinguishes between these two types of trajectories, achieving a VOC-F1 of 0.89 on successful videos and only 0.44 on failed ones. This demonstrates VLAC's strong ability to identify erroneous actions. Furthermore, for RoboFAC, the model trained with Ego4D data shows an even greater gap between successful and failed videos, indicating that human video data provides significant benefits for embodied task process understanding. The specific experimental results are shown in the Appendix D Table 3.

We show some example results in Figure 4, where (b–h) illustrate performance on datasets across different entities, scenes, and tasks, demonstrating the strong generalization capabilities of the proposed model. More specifically, Figure 4 (b) highlights the model's understanding of the human dataset. Figures 4 (c) and (d) show zero-shot and one-shot results for the same process, illustrating how in-context learning enhances the model's ability to comprehend new task processes. Figures 4 (e) and (g) depict successful task processes, while Figures 4 (f) and (h) present the corresponding failed task processes, indicating that the model can clearly distinguish between successful and unsuccessful processes. These examples show our model can robustly generalize across heterogeneous embodiments, scenes, and tasks, leveraging in-context learning to accurately contrast and predict fine-grained task progress, and even differentiate subtle failure modes from successful actions.

## 4.5 VLAC Actor Performance

| TASK | Open Cooker | | Pick and Place Bowl | | Unfold Mat | | Desktop Sweep | | Rice Transfer | | **Average** | |
|---|---|---|---|---|---|---|---|---|---|---|---|---|
| Metrics | task progress | success rate | task progress | success rate | task progress | success rate | task progress | success rate | task progress | success rate | task progress | success rate |
| $\pi_0$ | 85% | 85% | 54% | 40% | 32% | 0% | 30% | 10% | 45% | 30% | 49.2% | 27% |
| $\pi_0$ +Lighting Transfer | 0% | 0% | 7.5% | 0% | 35.5% | 5% | 31.5% | 10% | 9% | 0% | 16.7% | 3% |
| VLAC w/o pretrain | 50% | 50% | 0% | 0% | 2.5% | 0% | 0% | 0% | 56% | 30% | 21.7% | 16% |
| VLAC+FAST | 80% | 80% | 85% | 85% | 72.5% | 55% | 65% | 50% | 74.5% | 60% | 75.4% | 66% |
| VLAC | 90% | 90% | 85% | 85% | 91% | 85% | 62.5% | 40% | 84.5% | 75% | **82.5%** | **75%** |
| VLAC+Lighting Transfer | 85% | 85% | 65% | 65% | 72.5% | 60% | 49.5% | 25% | 75.5% | 50% | 69.5% | 57% |
| VLAC+Scene Transfer | 90% | 90% | 70% | 65% | 80% | 70% | 60.0% | 40% | 68% | 50% | 73.6% | 63% |

Table 1: Performance of VLAC's action generation across scenarios.

To validate the action generation capability of VLAC, we collected 100 samples for each testing task and trained the 8B VLAC model across all tasks. Additionally, since autonomous evolution in the real world often requires exploring different environments independently, the model must possess strong generalization abilities to adapt to scene variations. To test this, we evaluated the VLAC model's success rate under lighting disturbances and scene changes without requiring extra data collection. As shown in Table 1, the "Lighting Transfer" scenario involved turning off fluorescent lights and using colored flashing light sources as disturbances. For the "Scene Transfer" scenario, tests were conducted on two different workbenches located at different sites and different settings, with varying camera perspectives that were not precisely calibrated. The examples of training data and evaluation environments can be seen in Appendix Figure 6. The results show that $\pi_0$ Black et al. (2024), a typically state-of-the-art manipulation model, attains limited gains from cross-scene small dataset fine-tuning and is less robust to perturbations. In contrast, our model (VLAC) demonstrates stronger robust generalization, consistently generating reasonable actions even under extreme lighting changes, making it well-suited to dynamic and unpredictable environments. Meanwhile, "VLAC w/o pretrain" refers to the model without task progress pretraining, which leads to a significant drop in success rate. Although the actions generated during testing remain reasonable, the model struggles to accurately determine the current task state. For example, in the pick-and-place task, it may proceed to the placement step even if it has not successfully grasped the object. Our process understanding task enhances the model's ability to interpret the progression of tasks depicted in images, thereby ensuring a higher success rate at each execution stage. "VLAC+FAST" utilizes the FAST action tokenizer, which requires new semantic action mapping and offers limited leverage of process understanding in semantic space, resulting in a slightly lower average success rate compared to direct semantic actions. However, the success rate of tasks varies depending on the difficulty of the tasks, and performance is often partially lost during transfer. To improve the success rate during real-world deployment, further autonomous learning and evolution in actual environments and tasks can be assisted with a critic.

## 4.6 REAL-WORLD RL RESULTS

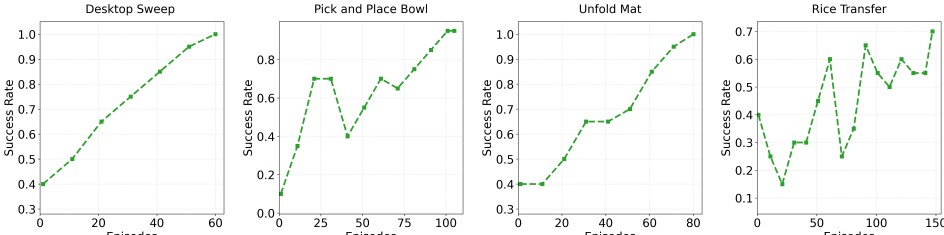

Figure 5: Experimental results of real-world reinforcement learning on four tasks.

In this subsection, we primarily investigate the effectiveness of real-world reinforcement learning when using VLAC as a dense reward model. To thoroughly validate the model's autonomous improvement capability, we reduced the number of imitation learning samples, enabling the agent to start learning in the real world from a relatively low initial performance. As shown in Figure 5, the success rate for each data point is calculated over 20 episodes, and our VLAC lifts the success rate from about 30% to approximately 90% within 200 real-world interaction episodes.

## 5 CONCLUSION

In this work, we introduced VLAC, a generalist Vision-Language-Action-Critic model that unifies action generation and task progress understanding within a single autoregressive architecture. By leveraging a scalable pair-wise progress prediction approach, VLAC provides dense and precise feedback signals, enabling robust self-improvement in dynamic and open-ended real-world environments. The model is trained on large-scale, diverse datasets, integrating both human and robot data, and demonstrates strong generalization capabilities across tasks, scenes, and entities. Extensive experiments show that VLAC not only discriminates between successful and failed trajectories with high accuracy but also generates reliable actions under challenging conditions, such as lighting disturbances and scene changes. Furthermore, VLAC's integration with reinforcement learning enables autonomous exploration and significant improvement in task success rates. Overall, VLAC establishes a unified and scalable framework for advancing robotic perception, manipulation, and autonomous learning in complex real-world settings.

## 6 REPRODUCIBILITY STATEMENT

To ensure the reproducibility of our work, we have submitted the code as supplementary material and will publicly release the code, fine-tuning dataset, and some pre-trained VLAC models to the public once our paper is accepted.

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

# A    MULTI-SOURCE TRAIN DATASETS

| Dataset Name | Size/samples | Size/h | Tasks | Mixture Weight |
|---|---|---|---|---|
| Ego4D HOD Pei et al. (2025) | 157M | 3k | TPU | 14.6% |
| InstructPix2Pix Brooks et al. (2023) | 36k | - | GVL | 0.4% |
| RobotVQA Sermanet et al. (2024) | 50k | - | GVL | 0.6% |
| Spot the diff Jhamtani & Berg-Kirkpatrick (2018) | 27k | - | GVL | 0.3% |
| Llava Liu et al. (2023) | 633k | - | GVL | 4.7% |
| SpatialQA Chen et al. (2024) | 781k | - | GVL | 5.8% |
| AGIBOT Bu et al. (2025) | 8M | 73 | TPU,GVL,VLA | 3.0% |
| Bridge Walke et al. (2023) | 2M | 135 | TPU,VLA | 9.1% |
| Droid Khazatsky et al. (2024) | 40M | 741 | TPU,VLA | 30.0% |
| FMB Luo et al. (2025) | 2m | 144 | TPU,VLA | 9.7% |
| RoboSet Bharadhwaj et al. (2024) | 4m | 130 | TPU,VLA | 17.4% |
| Self Collected | 946k | 18 | TPU,VLA | 4.4% |

**Note:** TPU = Task Progress Understanding; GVL = General Vision-Language; VLA = Vision-Language-Action.

Table 2: Overview of Data Mixture.

We collected and processed data from various sources, including human demonstration data, multiple types of robotic arm data, and VQA datasets. In addition, we collected a small portion of our data specifically for fine-tuning action representations on our robotic arm. The details of the datasets and their combinations are shown in Table 2. In total, we sampled 40 million data points (some of which include multi-turn dialogues) for training.

# B    EVALUATION METRICS

Evaluation Metrics. Following GVL Ma et al. (2024), we use Value-Order Correlation (VOC) as the primary metric for task progress understanding. VOC is the rank correlation between predicted step values and their chronological order in an expert video; higher values (-1 to 1) indicate monotonic task progression. To assess the robustness of pairwise methods, we introduce Value-Reversed-Order Correlation (VROC), computed after reversing the sequence; performance is more stable when VROC is close to VOC. We also report VOC-F1, the harmonic mean of VOC and VROC, to jointly capture forward and reversed consistency. Negative Rate (NR) is the fraction of pairwise comparisons whose predicted progress difference is negative, reflecting non-contributory or regressive actions. For action generation, following $\pi_{0.5}$ Intelligence et al. (2025), we measure success rate and human-rated task progress, the latter enabling finer-grained evaluation when tasks are partially completed. In real-world RL experiments, we focus on success rate, reported as the moving average over the most recent 20 episodes.

Following the GVL Ma et al. (2024), we use Value-Order Correlation (VOC) as the main evaluation metric for task progress understanding. This metric computes the rank correlation between the predicted values and the chronological order of the input expert video:

$$\begin{cases} \text{VOC} = \text{rank-correlation}(\text{argsort}(v_1, \ldots, v_T); \text{arange}(T)) \\ v_i = v_{i-\Delta t} + c_{i-\Delta t, i}(1 - v_{i-\Delta t}) \\ v_0 = 0 \end{cases} \tag{6}$$

VOC ranges from -1 to 1. Higher VOC scores indicate better task completion, with task progression increasing over time. A good critic model should achieve high VOC scores when evaluated on expert videos. To better evaluate pair-wise methods, we additionally construct Value-Reversed-Order Correlation (VROC). During testing, the entire sequence is reversed, so the order of pairwise frames is inverted and the predicted values should also be reversed. The closer the VROC score is to the VOC score, the better and more stable the model's performance. We further define VOC-F1 as:

$$\text{VOC-F1} = 2 \cdot \frac{\text{VOC} \cdot \text{VROC}}{\text{VOC} + \text{VROC}} \tag{7}$$

which comprehensively evaluates the correlation of task progression and temporal order in the video. If $\text{VOC} \cdot \text{VROC} < 0$, set VOC-F1 = 0. Additionally, to evaluate the fine-grained performance of

the critic model, we use Negative Rate (NR), which measures the proportion of reversed process pairs:

$$\text{NR} = \frac{N(c_{i,i+\Delta t} < 0)}{N} \tag{8}$$

where $N(c_{i,i+\Delta t} < 0)$ is the number of negative evaluations, and $N$ is the total number of evaluations. This metric reflects how many actions in the video do not contribute to task progression and is also an important indicator of the quality of a trajectory.

Following the $\pi_{0.5}$ Intelligence et al. (2025), we success rate and task progress as evaluation metrics for action generation. Task progress is evaluated by humans, allowing for more detailed progress assessments when the task is not fully completed, and providing a finer evaluation of the model's manipulation ability.

For real-world RL experiments, we mainly investigate the success rate. The success rate reflects the capability of the policy and it is evaluated the average success rate of the past 20 episodes.

## C  EVALAUATION TASKS

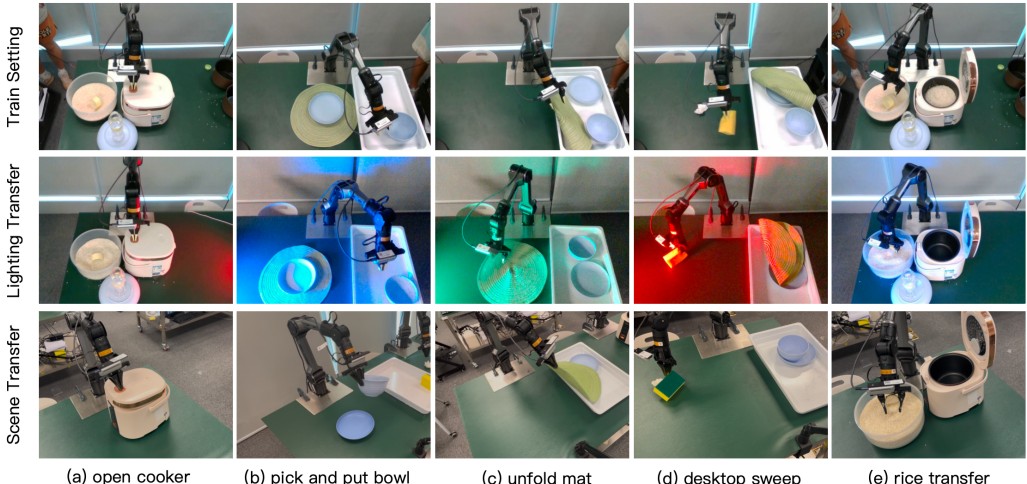

Figure 6: Illustrations of scene/lighting transfer.

We design five manipulation tasks in various environments as shown in Figure 6. The **Lighting Transfer** scenario involved turning off fluorescent lights and using colored flashing light sources as disturbances. For the **Scene Transfer** scenario, tests were conducted on two different workbenches located at different sites and different settings, with varying camera perspectives that were not precisely calibrated. These tasks are expected to constitute a rich and relatively complete kitchen scene from cooking food to setting up the table. In addition, these manipulation tasks are diverse from two perspectives: 1) manipulating different types of objects, i.e., rigid objects (a,b), flexible objects (c,d), and granular objects (e); 2) requires different manipulation ability, either precise goal reaching, i.e., touch/push objects (a,d), or grasp proper parts (b,c), or dynamic manipulation (e).

**(a) Open Cooker** In this task, the robot is supposed to press the button on top of the cooker. This task is considered successful only if the cooker lid is opened. To achieve this goal, the robot should go down and press the button with proper force. If the force is too great, the cooker or the arm may be damaged; if the force is too small the cooker can not be opened.

**(b) Pick and Place Bowl** In this task, a plate is placed on the table and a bowl is placed in a tray, and the robot is supposed to pick up the bowl and place it on the plate. The task is considered successful if the bowl is properly put on the plate. This task requires precise and gentle gripping of the bowl's rim and delivering it precisely to the center of the plate.

**(c) Unfold Mat** In this task, a mat is folded in the tray, and the robot is supposed to grab the mat, lift it up, and then release it to unfold the mat. This task is considered successful if the mat unfolds well

on the table. The difficulty lies in two points: 1) the robot must grab a proper part of the mat; 2) the mat must be raised high enough so that it can spread naturally, otherwise it may still be folded after falling on the table.

**(d) Desktop Sweep Disposal** In this task, there is a white trash on the table and the robot is supposed to pick up the scrub sponge and sweep trash into the trash can. The task is considered successful if the trash is swept into the nearby can. Similar to task (A), this task requires the robot to precisely reach the trash and push it with a proper force. Too much or too little force can lead to failure. This task is also named **Desktop Sweep** for abbreviation.

**(e) Rice Scooping and Transfer** In this task, the robot is assumed to firstly scoop a spoonful of rice from the jar, and then pour the rice into the cooker. The task is considered successful if the rice is transferred from the jar to cooker without spilling. The difficulty lies in the granular objects are not easy to be obtained and the transportation also requires the robot to be very stable. This task is also named **Rice Transfer** for abbreviation.

## D    DETAILS OF EXPERIMENT RESULTS

Detailed results of the experiments are summarized in the Table 3 on the next page. The table includes metrics such as VOC, VROC, and NR scores.

## E    THE USE OF LARGE LANGUAGE MODELS (LLMS)

We only utilize large language models for polishing the writing.

| Dataset | GVL (Gemini-1.5-Pro) | VLAC-8b w/o pair-wise | VLAC-8b w/o Ego | | | | | | VLAC-8b | | | | | | VLAC-2b | | | | | |
|---|---|---|---|---|---|---|---|---|---|---|---|---|---|---|---|---|---|---|---|---|
| | | | zero-shot | | | one-shot | | | zero-shot | | | one-shot | | | zero-shot | | | one-shot | | |
| | VOC | VOC | VOC | VROC | NR | VOC | VROC | NR | VOC | VROC | NR | VOC | VROC | NR | VOC | VROC | NR | VOC | VROC | NR |
| RT1 | 0.74 | 0.44 | 0.87 | 0.77 | 0.19 | 0.96 | 0.95 | 0.11 | 0.71 | 0.72 | 0.22 | 0.91 | 0.95 | 0.14 | 0.69 | 0.49 | 0.22 | 0.80 | 0.86 | 0.25 |
| Bridge | 0.51 | 0.67 | 0.96 | 0.96 | 0.05 | 1.00 | 1.00 | 0.00 | 0.97 | 0.99 | 0.02 | 1.00 | 1.00 | 0.00 | 0.94 | 0.95 | 0.08 | 0.98 | 0.99 | 0.05 |
| DROID | -0.01 | 0.50 | 0.85 | 0.86 | 0.09 | 0.96 | 0.95 | 0.06 | 0.92 | 0.94 | 0.04 | 0.93 | 0.95 | 0.06 | 0.75 | 0.72 | 0.13 | 0.79 | 0.79 | 0.11 |
| RoboNet | -0.85 | 0.03 | 0.20 | 0.32 | 0.50 | 0.76 | 0.71 | 0.23 | 0.00 | 0.00 | 0.00 | 0.59 | 0.76 | 0.31 | -0.04 | 0.00 | 0.00 | -0.04 | 0.81 | 0.52 |
| Dobb-E | 0.53 | 0.26 | 0.49 | 0.42 | 0.39 | 0.75 | 0.65 | 0.28 | 0.47 | 0.55 | 0.34 | 0.74 | 0.78 | 0.26 | 0.03 | 0.43 | 0.36 | 0.37 | 0.55 | 0.31 |
| RH20T | none | 0.15 | 0.24 | 0.19 | 0.34 | 0.68 | 0.65 | 0.21 | 0.17 | 0.34 | 0.32 | 0.64 | 0.77 | 0.22 | 0.49 | 0.40 | 0.32 | 0.49 | 0.78 | 0.25 |
| EgoDex | none | 0.05 | 0.58 | -0.41 | 0.29 | 0.64 | 0.36 | 0.32 | 0.05 | 0.48 | 0.48 | 0.44 | 0.78 | 0.38 | 0.00 | 0.59 | 0.47 | 0.57 | 0.69 | 0.36 |
| RoboFAC-success | none | 0.57 | 0.700 | 0.39 | 0.26 | 0.86 | 0.87 | 0.21 | 0.76 | 0.59 | 0.14 | 0.83 | 0.95 | 0.20 | 0.46 | 0.69 | 0.20 | 0.73 | 0.92 | 0.29 |
| RoboFAC-fail | none | 0.18 | 0.45 | 0.16 | 0.34 | 0.66 | 0.56 | 0.32 | 0.30 | -0.06 | 0.22 | 0.41 | 0.47 | 0.35 | -0.05 | 0.34 | 0.26 | 0.19 | 0.57 | 0.43 |

Table 3: Performance of VLAC's task progressing understanding across entities, scenarios, and tasks.