# OpenReview forum: "VLAC: A Generalist Action-Critic Model via Pair-wise Progress Understanding"
_ICLR.cc/2026/Conference — ICLR 2026 Conference Withdrawn Submission_

### Official Review · Reviewer_fPty · 2025-10-24

**Soundness:** 2
**Presentation:** 2
**Contribution:** 2
**Rating:** 2
**Confidence:** 4

**Summary:**

This paper proposes VLAC, a unified Vision–Language–Action–Critic architecture that aims to predict both robot actions and progress toward task completion. The method uses a pairwise progress-understanding mechanism: given two frames from a video, the model predicts their relative task progress and uses this signal as an intrinsic reward for reinforcement learning. The model is trained on a large mixture of web-scale video, human demonstration, and robot datasets. Experiments claim that the learned critic can generalize across several benchmarks and that using its reward signal helps improve on-robot reinforcement learning efficiency and success rate.

**Strengths:**

•	The topic—learning dense progress and reward signals from visual data for RL—is relevant to embodied intelligence and of potential long-term interest.

•	The model attempts to unify vision, language, and action learning under a shared architecture, which aligns with the trend of generalist agent design.

**Weaknesses:**

•	The pairwise progress prediction and intrinsic reward learning are straightforward extensions of established ideas in temporal contrastive learning and visual reward modeling. There is little conceptual or methodological innovation beyond scaling existing components.

•	Key mechanisms—such as how progress labels are computed, how critic and actor losses interact, and how the model architecture fuses modalities—are described imprecisely. The paper lacks equations, algorithmic clarity, and ablation to demonstrate each part’s contribution.

•	Experimental results are not rigorously supported. No proper statistical tests or confidence intervals are reported, and baselines are weak or missing (e.g., comparison to existing learned reward or progress models). Claims of “generalization across unseen datasets” are not verified with strong quantitative metrics or external benchmarks.

•	The training pipeline depends on massive compute (200 A100 GPUs), and many details are hidden in appendices. There is no released code, no hyperparameter sweep, and insufficient description to allow independent reproduction.

•	The paper repeatedly claims strong generalization and practical real-world success, but given the weak evidence and missing baselines (e.g. ReinboT), these claims are overstated relative to the presented data.

•	The writing is unclear, especially in the methods section. The internal logical relationships between the various modules are difficult to understand. The experimental setup and result presentation are crude, making it difficult to convincingly demonstrate the effectiveness of the methods.

Ref: Zhang, H., Zhuang, Z., Zhao, H., Ding, P., Lu, H., & Wang, D. (2025). ReinboT: Amplifying Robot Visual-Language Manipulation with Reinforcement Learning. arXiv preprint arXiv:2505.07395.

**Questions:**

1.	How is the pairwise progress label precisely defined? Is it a normalized temporal difference or learned via contrastive supervision? Please provide mathematical formulation.

2.	How is the critic integrated into RL training—does it produce scalar rewards or dense time-series signals? How are they scaled relative to environment rewards?

3.	What are the quantitative baselines for the critic compared to existing visual progress estimators or learned reward models?

4.	What measures were taken to ensure that the critic does not produce misleading rewards in visually noisy or multi-goal environments?

5.	Can the reported real-robot improvements (≈30%→90%) be replicated with fewer compute resources or smaller models, or are they specific to large-scale pretraining?

---

> ### Author Response · Authors · 2025-12-02
>
> Thank you for the comments. In embodied intelligence, we introduce pair-wise supervision into critic construction and jointly train it with a VLA for the first time. This design naturally addresses two key issues in public datasets, namely the lack of negative examples and the absence of fine-grained labels, while fully leveraging human data. Our pair-wise method consistently sets the first image as the task start point to estimate progress, which avoids inconsistencies in the zero-progress reference commonly found in public data. Joint training with the VLA verifies that pair-wise progress estimation significantly improves VLA generalization and success rates. Its application in reinforcement learning further validates the effectiveness and accuracy of pair-wise progress estimation.
>
> 1. We provide the definitions in Equations (1) and (5). In the paragraph adjacent to Equation (1) on page 4, line 198, we present the formula for generating pair-wise progress labels with a range from −1 to +1. In the same paragraph, we also describe how image pairs are sampled from trajectories.
>
> $c_{i,i+Δt} = VLAC(o_i, o_{i+Δt}; l_task)$
>
> $c_{i,i+Δt} = Δt / (T − i)$
>
> 2. In reinforcement learning, based on the estimation progress, the critic outputs dense reward signals and we update the policy using PPO. Because our experiments are conducted in the real world, task completion is judged by our critic model, and the environment reward is the critic’s estimate of task progress.
>
> 3. Our primary focus is on generalization and accuracy, enabling application to new scenes and tasks without training at test time. We mainly compare against GVL and the single-point estimation method trained on the same dataset to validate the effectiveness of the pair-wise approach.
>
> 4. We mainly use in-context learning to enhance robustness by incorporating references from the current environment as context in the inputs. In addition, during reinforcement learning, we get reward when progress surpasses thresholds, which further improves the accuracy of the supervisory signal.
>
> 5. The real-robot performance improvements can be achieved with modest compute, requiring only a single A100 GPU during the reinforcement learning phase. However, large-scale pretraining is essential to ensure the critic model’s generality. We plan to release the pretrained model to provide a strong foundation for cross-scene applications and for further finetuning of the critic. We have also included training and inference code in the materials along with required parameter settings to facilitate reproducibility.

---

### Official Review · Reviewer_bypM · 2025-10-30

**Soundness:** 2
**Presentation:** 3
**Contribution:** 1
**Rating:** 2
**Confidence:** 4

**Summary:**

The paper proposes VLAC, a VLA model which also predicts the number of frames between subsequent images. This is used as a goal-conditioned critic function. The authors

**Strengths:**

* Though not new, the idea of using a goal representation (or second image) for progress estimation is good.
* The authors provide fairly extensive detail on all of the different training data they use for the VLAC model.

**Weaknesses:**

To me, it seems like the authors have a) proposed a goal-conditioned version of GVL and b) trained a model for GVL instead of using an off-the-shelf model (which was the purpose of GVL -- no training required). However, these choices do not appear extremely well justified at present.
* The point of GVL is that it was training free -- you could just prompt an existing model to get value estimates. In that regard it doesn't make sense to have GVL be the main baseline here. The authors should be comparing to other approaches or other ways of training a value function (particularly other ways of training a goal conditioned value function)
* The value-function based evaluation is fairly weak. Only VOC is used (which I think is a rather weak metric as maximizing VOC is only meaningful if the data is actually from a true expert, which it rarely is). Why not consider policy learning on filtered data etc.?
* The authors call their method "pair-wise progress estimation" when it really seems like a goal conditioned value function $V(s,g)$ instead of just $V(s)$. This makes it seem like "pair-wise proress estimation" is more of a new idea than I think it really is. The authors could do a better job contextualizing their method. There have been several works that have tried to train large scale goal conditioned value functions. For example, ViVa: Video-Trained Value Functions for Guiding Online RL from Diverse Data.
* the connection between the policy learning experiments and the story is a bit unclear. Is VLAC a critic or is it a VLA? Is the point that by being a critic and an actor at the same time you get better performance? This is a bit clear. One could evaluation the critic performance by filtering data according to the value function. One would show that doing both is better than doing just one by doing careful ablations -- but that isn't clear. What are the policy learning results trying to show in the context of the story?
* there are no baselines for RL experiments.
* I think training a model is useful and cool! but it is not necessarily a scientific contribution on its own -- what constitutes a contribution are the lessons learned, discoveries made, resources provided, new methods etc. While the paper does a good job detailing what the VLAC model is and how it performs, what the scientific contribution is remains a bit unclear to me.

Nits:
* The authors frequently complain about "single-point estimates" but this term isn't well defined. I would usually expect it to indicate a single-sample estimate, but I think the authors are implying that they use an image pair instead of one image. However think that these are different modeling assumptions, since neither approach is actually using multiple independent samples.

**Questions:**

* How is VOC computed on the dataset given the model is "goal-conditioned"? Do you compute the VOC by using subsequent frames and summing up the differences? Do you compute it by comparing each frame with the final frame? this is important because it can determine what VLAC is good at -- maybe its really good at seeing if there is progress in neighboring frames but worse at evaluation between distant frames.
* If the authors are going to train a model from scratch, why did they not compare to other approaches that train from scratch? E.g. learning a goal conditioned value function with TD, latent intentions, VIP, etc. These seem like more apt baselines than GVL, which was intended as a training-free method.
* Is there contamination between the VLAC pretraining data and the evaluation tasks? Are they more indistribution than the pi0 pretraining mixture?

---

> ### Author Response · Authors · 2025-12-02
>
> Thank you for the suggestions and for recognizing our model. Our pair-wise progress approach improves the generalization and stability of value models. As shown in Figure 3, it yields significant gains compared with both single-point methods (VLAC w/o pair-wise) and GVL. We also introduce joint training of the critic and the VLA actor, which enhances VLA generalization and accuracy. In addition, the pair-wise progress method assists real-world RL, increasing task completion rates. These results validate the effectiveness of pair-wise progress across multiple scenarios and constitute our primary scientific contribution.
>
> 1. Due to space limits, the VOC computation is provided in the supplemental material as Equations (6) and (7). In Appendix, it is based on pair-wise critic that are progressively accumulated. Because the inversion process is correspondingly reversible with the pair-wise label generation procedure, long-horizon effects are preserved. Figure 4 also demonstrates performance on long-range comparisons.
>
> 2. Our model is post training on the InternVL, followed by additional training on large-scale data. The trained model is designed to require no further training when transferred across tasks and scenes, aiming for strong generality. Therefore, GVL remains our preferred baseline. We evaluate on six datasets that differ substantially from our training data, and no training is performed in various test scenarios. Specifically, the datasets RT-1, RoboNet, Dobb-E, RH20T, EgoDex, and RoboFAC were never used during training; the evaluation is zero-shot and one-shot in-context, as shown in  Figure 3.
>
> 3. There is no contamination. We intentionally selected datasets that are highly dissimilar from the training distribution—different robot arms, different tasks, and different environments. Supplemental Table 2 lists all training data and their mixing ratios. The test datasets (RT-1, RoboNet, Dobb-E, RH20T, EgoDex, and RoboFAC) were not used in training in any form.

---

### Official Review · Reviewer_chZw · 2025-11-02

**Soundness:** 2
**Presentation:** 1
**Contribution:** 2
**Rating:** 2
**Confidence:** 4

**Summary:**

This paper proposes a generalist vision-language-action model that unifies action generation and task progress understanding within a single autoregressive architecture. Specifically, the authors introduce a pairwise progress understanding approach that predicts the task progress delta between any two images in a visual trajectory, while generating the corresponding action based on the first image. The model is trained using a combination of large-scale, multi-source human data (without explicit action annotations) and robot data (with action information), while also incorporating general vision-language data to enhance its world knowledge. The proposed framework’s progress understanding capabilities are evaluated across eight diverse datasets, demonstrating strong generalization to new tasks and environments. Furthermore, the authors validate the model’s effectiveness through real-world manipulation experiments.

**Strengths:**

The authors investigate an important and timely direction, i.e., progress understanding for real-world manipulation tasks. They propose a pairwise progress understanding approach that predicts the task progress delta between any two images within a visual trajectory and generates the corresponding action based on the first image. The proposed method demonstrates strong performance across eight public datasets, including six unseen during training, highlighting the model’s impressive generalization capability.

**Weaknesses:**

Overall, the proposed framework addresses a timely and important direction. However, I have several concerns regarding the overall presentation of the manuscript:

1. **Motivation:** In the first paragraph, the authors argue that existing VLA models fail to generalize across scenes with robustness due to the lack of efficient feedback. I find this claim overstated. There are multiple factors that hinder the generalization ability of VLA models. The authors are encouraged to tone down this statement and clarify that the direction explored in this manuscript represents one of several possible approaches to improving generalization.
2. **Pairwise progress comparison is not a new idea:** The concept of progress understanding through pairwise comparison has been explored in prior works (e.g., [1–3]). However, the manuscript does not discuss how the proposed approach differs from these existing studies. Moreover, the paper lacks a discussion of the specific challenges encountered when incorporating this idea into a VLA framework, which makes the contribution appear less insightful and more like a system implementation paper.

[1] Hung et al., VICtoR: Learning Hierarchical Vision-Instruction Correlation Rewards for Long-horizon Manipulation, ICLR 2024

[2] Xue et al.,, Progress-Aware Video Frame Captioning, CVPR 2025

[3] Kung et al., What Changed and What Could Have Changed? State-Change Counterfactuals for Procedure-Aware Video Representation Learning, ICCV 2025.

3. **Is the proposed pairwise progress comparison the main contributor to progress understanding?** From Figure 3, it appears that this component contributes to performance improvements on datasets such as RT-1, Bridge, DROID, and Dobb-E. However, the improvement is not consistent across datasets. In particular, RoboFAC shows a significant performance drop when using pairwise comparison, suggesting that the module may be ineffective in estimating progress in certain scenarios. Although performance improves when additional information is incorporated, this inconsistency raises concerns about the true contribution of the pairwise comparison module.
4. **What is the impact of pairwise progression on action generation?** A key follow-up question is whether pairwise progression meaningfully contributes to real-world robotic manipulation. The results presented in Table 1 do not clearly convey this point, making it difficult to assess the practical importance of the proposed mechanism for action generation.
5. **Limitations in ablation studies for real-world manipulation:** The current evaluation and demonstrations are insufficient to validate the effectiveness of the proposed framework in real-world manipulation tasks. Specifically, the paper lacks ablation studies comparing model performance with and without the pairwise progress understanding module. Given that the overall pipeline involves multiple components, including diverse datasets and design choices, it is challenging to isolate and attribute performance gains specifically to the proposed progress understanding approach.
6. **Lack of empirical results for progress estimation in real-world robotic manipulation:** The current demonstrations are entirely static (e.g., Figures 4 and 6), making it difficult to justify the framework’s effectiveness in real-world manipulation scenarios.

**Additional Notes:**

1. The overall writing quality of the manuscript requires significant improvement. For example:
    - **L44–46:** This segment is not a complete sentence and should be revised for grammatical correctness.
    - **Equations (1)–(3):** The notations are not rigorous. In particular, the same function is presented with different argument formats, which is conceptually inconsistent and potentially confusing.

**Questions:**

1. Motivation: Do the authors provide sufficient justification for their claim that the lack of efficient feedback is the primary reason existing VLA models fail to generalize across scenes? Could this claim be overstated given that multiple factors affect generalization, and should the authors clarify that their proposed direction is only one of several possible approaches?

2. Novelty of the pairwise progress comparison: How does the proposed pairwise progress comparison approach differ from prior works such as Hung et al. (ICLR 2024), Xue et al. (CVPR 2025), and Kung et al. (ICCV 2025)? What unique challenges or insights arise when incorporating this idea into a VLA framework?

3. Contribution of the pairwise progress comparison module: Is the proposed pairwise progress comparison truly the main contributor to progress understanding? Given that performance improvements are inconsistent across datasets (e.g., performance drops on RoboFAC), how do the authors explain these variations and assess the module’s overall effectiveness?

4. Effect on action generation: To what extent does pairwise progression improve action generation performance, particularly in real-world robotic manipulation? The results in Table 1 do not make this relationship clear—can the authors provide more direct evidence or analysis?

5. Ablation studies: Why are there no ablation studies comparing model performance with and without the pairwise progress understanding module? Considering the pipeline includes multiple components and datasets, how can one isolate the true contribution of this module to overall performance?

6. Empirical validation in real-world settings: How effective is the proposed framework for progress estimation in real-world robotic manipulation? Since the demonstrations presented (e.g., Figures 4 and 6) are static, can the authors provide dynamic or real-robot experiments to support their claims?

---

> ### Author Response · Authors · 2025-12-02
>
> 1. Thank you for the comments. Indeed, generalization is influenced by multiple factors. Our approach is positioned as a self-evolving, data-driven method that enhances real-world generalization, with strong adaptability to dynamically changing environments.
>
> 2. Our method is not a video-understanding solution; it targets estimating task completion progress from a pair of images. The pair-wise design improves sensitivity to subtle state changes and yields a smoother progress curve. We also demonstrate strong generality with solid results on multiple OOD datasets. Moreover, we jointly train the critic and actor to enhance VLA robustness, and ablations verify the effectiveness of process understanding.
>
> 3. The performance drop on RoboFAC is actually indicative of the method’s discriminative power. RoboFAC has separate success and failure splits: we achieve a high VOC-F1 of 0.89 on the success split and 0.44 on the failure split. This contrast shows the model can clearly distinguish success from failure. Note that RoboFAC is a new dataset on which we only perform testing without training.
>
> 4. In Table 1, “VLAC w/o pretrain” corresponds to training a VLA without the pair-wise progress task. Its comparison against VLAC highlights the importance of process understanding. Furthermore, the subsequent reinforcement learning stage builds on VLAC’s pair-wise progress estimation, enabling autonomous improvement in the environment.
>
> 5. Table 1 includes relevant ablations, and we will further clarify their descriptions to make the contributions and isolation of effects more explicit.
>
> 6. Yes. While Figure 4 aims to illustrate overall judgments under different conditions, the same capability can be realized in real time. Dynamic materials are difficult to present directly in the paper; however, our project homepage will provide extensive demos, trial access to the model, and we will also release the code.

---

### Note · Authors · 2026-01-12

I have read and agree with the venue's withdrawal policy on behalf of myself and my co-authors.